# Earlier emergence of a temperature response to mitigation by filtering annual variability

B. H. Samset [1✉], C. Zhou [2], J. S. Fuglestvedt [1], M. T. Lund [1], J. Marotzke [3] & M. D. Zelinka [4]

The rate of global surface warming is crucial for tracking progress towards global climate targets, but is strongly influenced by interannual-to-decadal variability, which precludes rapid detection of the temperature response to emission mitigation. Here we use a physics based Green's function approach to filter out modulations to global mean surface temperature from sea-surface temperature (SST) patterns, and show that it results in an earlier emergence of a response to strong emissions mitigation. For observed temperatures, we find a filtered 2011–2020 surface warming rate of 0.24 °C per decade, consistent with long-term trends. Unfiltered observations show 0.35 °C per decade, partly due to the El Nino of 2015–2016. Pattern filtered warming rates can become a strong tool for the climate community to inform policy makers and stakeholder communities about the ongoing and expected climate responses to emission reductions, provided an effort is made to improve and validate standardized Green's functions.

[1] CICERO Center for International Climate Research, Oslo, Norway. [2] Nanjing University, Nanjing, China. [3] Max Planck Institute for Meteorology, Hamburg, Germany and Center for Earth System Research and Sustainability, Universität Hamburg, Hamburg, Germany. [4] Lawrence Livermore National Laboratory, Livermore, USA. ✉email: b.h.samset@cicero.oslo.no

The global, annual mean surface air temperature anomaly (GSTA) is arguably the most visible climate observable in public and policy debates. Since the 1970s, GSTA has increased at a rate of ~0.2 °C per decade, predominantly due to anthropogenic climate forcing[1,2]. However, due to internal variability in the climate system on annual-to-decadal timescales, warming rates calculated over shorter periods can vary strongly[3–7]. Over the last 50 years, decadal warming rates range from 0.0 to 0.4 °C[8]. The so-called "global warming hiatus" period, following the strong El Nino of 1998, is an example of a decade with low warming rate[9], whereas the most recent decade (2011–2020) has seen warming well above the multi-decadal average[8]. Consequently, climate scientists normally work with quantities computed over 30 years or longer, in order to reduce such influence of internal variability on their results.

The present situation is one of rapid global warming, combined with an increasing focus on emission reductions aimed at limiting surface temperature change to well below 2 °C above preindustrial levels, as envisaged by the Paris Agreement. Hence, it is critically important to establish with scientific certainty that reductions are having an observable effect on the global climate system[3,4]. Recent literature has shown that while strong mitigation relative to high emission scenarios, starting from 2020, is likely to have a measurable influence on 20-year warming rates by 2040[10,11], 10-year rate changes are unlikely to be discernable for several decades[3–5]. Warming rate changes resulting from weaker mitigation will also be progressively harder to observe.

The main reason for these difficulties is the strong influence on GSTA of interannual-to-decadal variability. A number of approaches exist that attempt to reduce this influence, either through estimating the underlying anthropogenic warming based on calculations of historical and current radiative forcing[12,13], that utilize the connection between global surface temperature and modes of variability such as the El Nino/Southern Oscillation (ENSO)[14], Atlantic and Pacific (multi-)decadal variability[15], or employ statistical techniques and pattern recognition methods to separate the anthropogenic warming rate from other influences (e.g., [16–24]). These approaches all have strengths and weaknesses. Some rely primarily on emission inventories and model estimates of the links between emissions, radiative forcing, and surface temperature responses, and on the assumption that we can identify model-derived modes of variability also in real-world observations. Others identify and subtract modes of variability based on observations alone, but do not include direct treatment of the physical connection between SST patterns and land temperature responses.

In this work, we present an approach to filtering interannual variability that is complementary to existing techniques, based around recently developed Green's Functions (GFs) that relate global mean responses in radiative fluxes, clouds, and surface temperature to local fluctuations in sea surface temperature[25–27]. Our aim is to quantify, as far as possible, the contribution to GSTA for a given year that can be related directly to responses from the realized SST pattern. The remaining surface temperature anomaly will then be a combination of internal variability on decadal and longer timescales (which is still present after our removal of the 10-year trend), other feedbacks and modulations (such as responses to warming patterns over land), and the underlying influence of anthropogenic global warming (notable patterns such as Arctic amplification).

## Results

### Green's Functions relate SST patterns to the GSTA. Given a pattern of sea surface temperature, GFs provide an estimate of the influence of that pattern on the global land surface temperature

anomaly for the given month or year. Here, we use a GF based on simulations with the CESM1 global climate model[26]. The underlying geophysical process is that a warmer sea surface will feedback on atmospheric temperatures, directly and through influencing evaporation, humidity, cloudiness, and circulation and that the strength of this modulation depends on the pattern of SST anomalies and the month in which it occurs. The ENSO is the most well-known example of this, but modulations of GSTA also arise from SST changes elsewhere. The GF method treats these in a consistent way, that also captures more than simply the influence of ENSO. As it provides the land surface air temperature response corresponding to an already known SST pattern, it is reasonable to think that this approach should be able to capture —and therefore filter out—a large component of internal variability. Also, although GFs are necessarily derived from a global climate model, the filtering approach relies only on the observed (and detrended) temperature pattern and is thus agnostic about the underlying pattern and magnitude of climate forcing mechanisms. This makes it complementary to the forcing-based approach mentioned above, and also to statistical approaches that do not retain information on physical processes linking modes of variability to temperature responses.

Figure 1 shows how the GFs are used to calculate the SST-induced influence on GSTA, for 3 example months. The observed GSTA (from HadCRUT5[8]) has first been detrended, via a 10-year moving boxcar average applied at each grid point (see Methods), to isolate (as far as possible) the influence of annual variability from decadal patterns and the effects of global warming since 1850. The resulting GSTA pattern is multiplied with the GF for the corresponding month, where the GF is a relation between an SST anomaly in a given grid point and the resulting influence on global mean surface temperature. Adding together the results for all ocean-dominated grid points yields the influence on GSTA (hereafter termed an SST modulation) from all parts of the global ocean, for that particular month. Note that we are here assuming that the modulations occur within the same month as the SST pattern; see below. Also, since the GF is (by construction) calculated from simulations with fixed sea surface temperatures, the SST modulation primarily includes land surface responses.

The application examples in Fig. 1 include a month at the height of the 1998 El Nino, where we find an SST modulation of GSTA of 0.3 °C; a month in 2010 where the SST variability pattern yielded no net modulation; and the most recent month of the dataset (December 2020) where weak La Nina conditions in the Pacific resulted in negative modulation of −0.1 °C.

**Filtering variability from recent GSTA observations**. Figure 2 shows the effect of GF-based filtering, i.e., subtracting the calculated SST modulations, on the global, annual mean temperature evolution in HadCRUT5 observations. The overall effect is to dampen strong deviations from the mean. The standard deviation of a residual relative to a 10-year running means is more than halved, from 0.092 °C to 0.041 °C. Note for instance the El Nino years of 1998 and 2016. Panel b shows the modulation for each year, and how it correlates with the Nino3.4 index. A perfect correlation is not expected, as ENSO is not the sole driver of GSTA variability, but we find a statistically significant co-variation (Pearson's $R = 0.56$ for 60 effective degrees of freedom, estimated via lag-1 autocorrelation[28]).

Panel c shows 10-year trends since 1850, calculated for the 10-year period ending in the given year. The observed trends (red) fall within ±0.4 °C/decade, with strong peaks around El Nino/La Nina years. The trends based on the modulation filtered GSTA (black) has a range of ±0.2 °C/decade, and exhibit much lower variability since 1970. We also show 30-year trends, ending in the

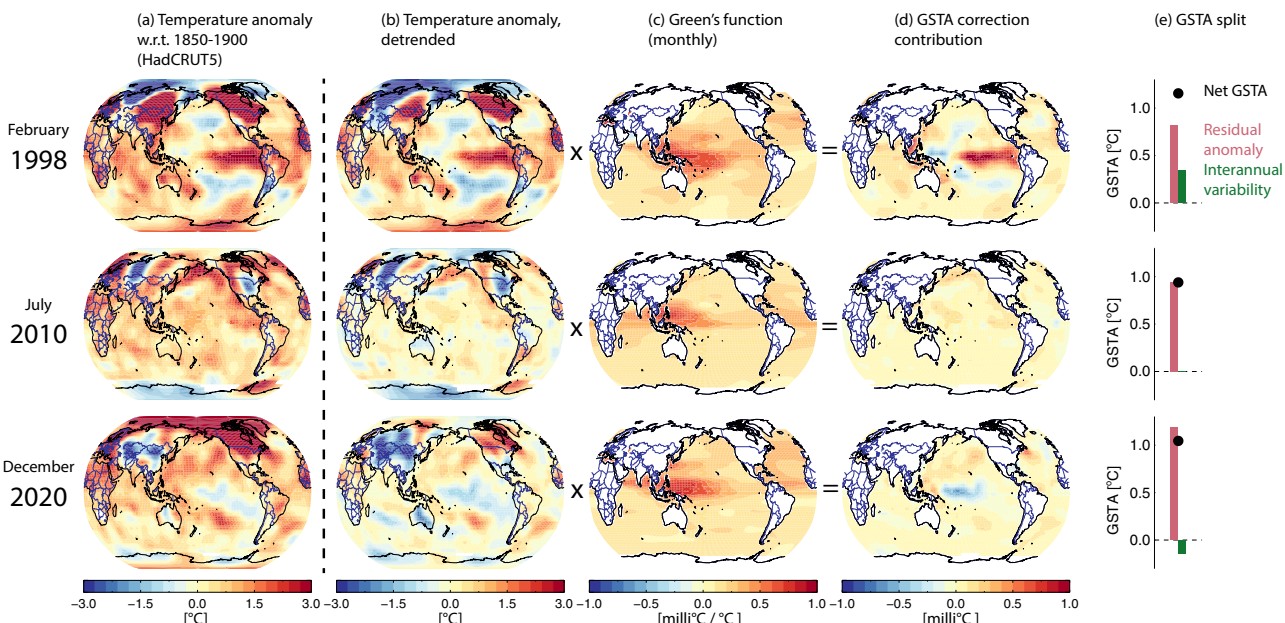

**Fig. 1 Calculation of modulations on global surface temperature anomaly (GSTA) from the pattern of sea surface temperatures (SST), for 3 example months. a** Raw temperature anomaly from HadCRUT5, relative to 1850–1900 mean for that month. **b** Detrended temperature anomaly. **c** Green's Function (GF) for the corresponding month, derived from the CESM1 global climate model, shows the contribution of an SST anomaly in a given grid cell to the GSTA anomaly. **d** GSTA modulation, found by multiplying the GF with the detrended anomaly map. **e** Net global surface temperature anomaly for that month, and its split into a variability (SST modulation) and a residual component. The variability (green bar) is equal to the sum of the map in column (**d**). The three months illustrate (top) strong El Nino conditions, (middle) neutral conditions, and (bottom) moderate La Nina conditions.

same year, and (as for panel a) show the residual standard deviation (RSD) of the 10-year filtered and unfiltered warming rates relative to this curve. While not as substantial as for individual years, there is still a marked decrease in the RSDs for the filtered 10-year trends, further strengthening the argument that GF-based filtering helps separate underlying trends in surface warming from the influence of internal variability.

For the most recent decade (2011–2020), unfiltered GSTA from HadCRUT5 shows a warming rate of 0.35 °C per decade. This is markedly higher than the 30-year (1991–2020) rate of 0.21 °C derived from the same dataset, partially due to the influence of weak La Nina conditions early in the period, followed by the strong El Nino of 2015–2016. The filtered GSTA yields a 2011–2020 warming rate of 0.24 °C per decade, and a 30-year trend that is indistinguishable from the unfiltered results (0.21 °C). Similarly, for the decade 2001–2010, which is in the center of the so-called global warming hiatus period, unfiltered HadCRUT5 observations yield a warming rate of 0.08 °C per decade, while the filtered results show 0.21 °C per decade, again similar to the most recent 30-year trend (1991–2020).

These results indicate that GF-based filtering of observed GSTA can give physically reasonable output and that it can be used as a transparent method for filtering out one component of interannual variability affecting decadal warming rates. Given this, we extend the approach to two ensembles of simulations using contemporary Earth System Models; a 10-member initial condition ensemble (MPI-ESM1.2[29]) and 61 realizations of historical and future GSTA from CMIP6[30,31]; and investigate the effect such filtering can have on the detectability of rate changes under different assumptions on future emissions. MPI-ESM1.2 was chosen because its representation of internal variability has been well studied in the previous work[6], it has an Equilibrium Climate Sensitivity consistent with recent assessment results[32,33], and provided 10 ensemble members (using identical emissions and other external conditions but representing different realizations of internal variability) for both

the historical period and for a range of future emission scenarios to CMIP6. A list of all CMIP6 models used is given in the Supplementary Information.

**Advancing the emergence time of mitigation.** In Fig. 3, we show the raw and GF filtered annual mean GSTA evolution from MPI-ESM1.2 (see Methods), using historical emissions and natural forcings (1850–2014) and four Shared Socioeconomic Pathways with widely varying future emissions and other anthropogenic activities (SSP1–2.6, SSP2–4.5, SSP3–7.0, SSP5–8.5[34–37]). Throughout the period 1850–2100, the GF calculations are able to filter out a component of interannual variability, leaving a smoother temporal evolution and reduced variability in decadal trends. The inset shows plumes of mean and ±1 standard deviation across the 10 ensemble members, for unfiltered and filtered calculations. Defining emergence as the year in which a 10-year running mean over the ensemble average response in the mitigated situation moves outside the plume for the unmitigated situation (see Methods), we find that for emissions following SSP1–2.6 relative to SSP5–8.5, the signal emerges from the noise 5 years earlier for filtered GSTA values than for unfiltered, i.e., in 2030 instead 2035. This indicates that the filtering method does indeed provide a way to more rapidly detect a change in the global mean warming rate resulting from differences in emissions pathways. For a 20-year running mean, the corresponding advancement of emergence is 3 years, reflecting the lower influence of internal variability on such longer-term means. Table 1 shows the corresponding shift in the year of emergence found for other combinations of assumed unmitigated and mitigated scenarios, for 10-year and 20-year means. In all cases, emergence moves earlier, by up to as much as 9 years. Performing the analysis using individual ensemble members as a signal, rather than the ensemble mean, shows similar advancement in the year of emergence after filtering, albeit with a spread of ~10 years in the actual year of emergence.

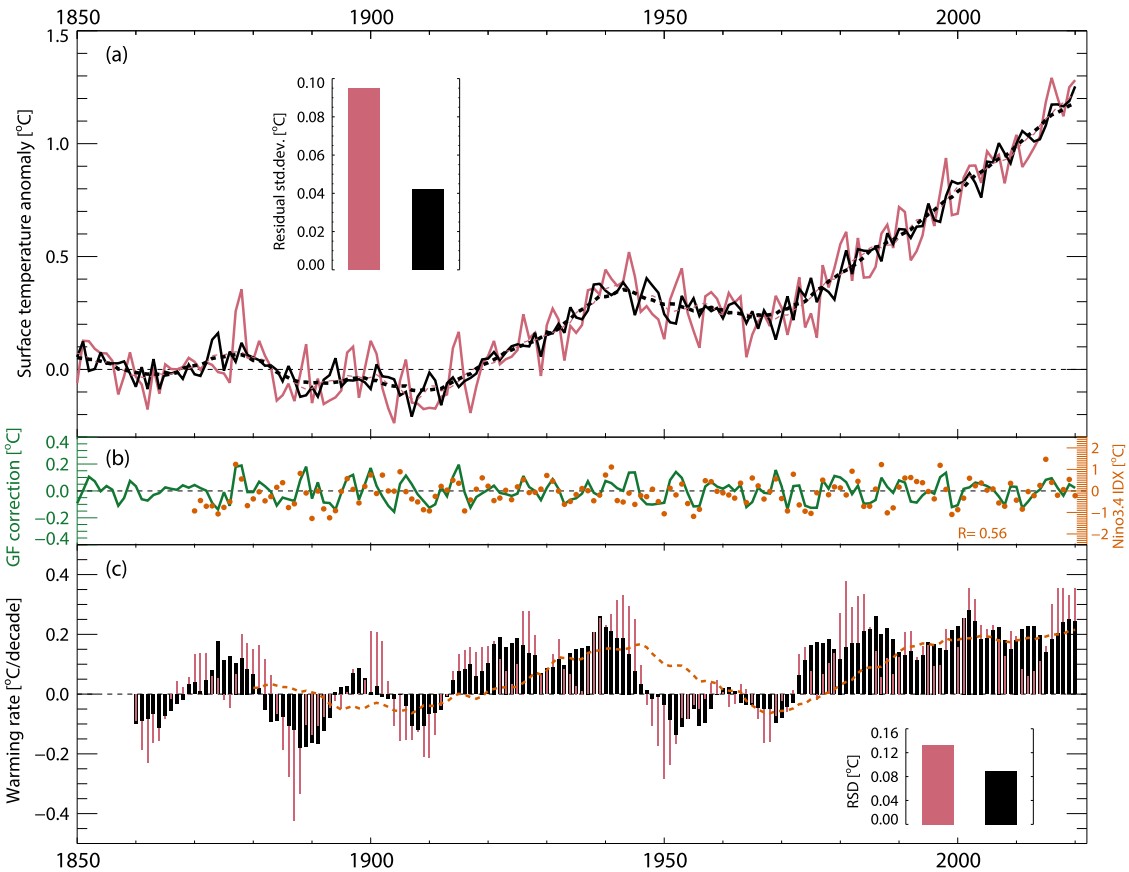

**Fig. 2 The effect of Green's function-based filtering on global annual mean surface temperature anomalies from HadCRUT5. a** Raw (red) and filtered (black) HadCRUT5 global mean surface temperature anomaly (GSTA) values, relative to 1850–1900. Inset: standard deviation of the residual to a 10-year boxcar average (dashed lines). **b** The annual mean modulation factors, and the Nino3.4 index for the corresponding year. **c** 10-year warming rates ending in the year shown, for raw and filtered HadCRUT5 GSTA values. The dashed line shows 30-year rates, ending in the same year. Inset: standard deviation of the residuals between the 10-year rates and the 30-year rates (as for the top panel).

The conclusion that filtering may allow for more rapid detection of the effects of mitigation is further strengthened by considering the effect it can have on the calculation of future warming rates. In Fig. 4, we first show the 10-year global warming rates for filtered and unfiltered GSTA evolution for the near-term periods of 2021–2030, 2031–2040, and 2041–2050, from MPI-ESM1-2. While there is still a marked spread between ensemble members after filtering, it is lower than for the unfiltered simulations. Even for 2021–2030, the warming rates from the filtered time series increase monotonically with the greenhouse gas emissions of the underlying scenario, which is not the case for the unfiltered time series. For 2041–2050 the SSP5–8.5 and SSP1–2.6 rates are fully separated, as are SSP3–7.0 and SSP1–2.6. Neither of these is true for the unfiltered situation. Except for SSP1–2.6 in 2041–2050, there are also no instances of negative 10-year warming rates in the filtered simulation results, whereas the unfiltered time series has multiple such cases. Figure 4 also shows 15-year and 20-year warming rates, for the period beginning in 2021. As expected, the longer the period of integration, the more similar the filtered and unfiltered rates become. There is, however, an appreciable reduction in the spread of rates in the filtered case, and improved separation between the scenarios, even for 20-year periods. For integrating periods at or above 20 years, warming rates calculated using filtered and unfiltered GSTA values are similar. This is as expected, as such calculations average out internal variability by construction. A further question is whether our results on the rapid separation between scenarios are sensitive to the model used, its

representation of variability, the underlying Equilibrium Climate Sensitivity, and other differences in the representation of the Earth System. To test this, we apply our filtering method to 61 realizations from the CMIP6 ensemble of simulations, where simulations are available from the same models for historical, SSP1–2.6, and SSP5–8.5 emissions. See Fig. 5, where we take as a starting point the most recent 30-year trend from each simulation (1991–2020), and check whether the trend is higher or lower for the subsequent 5, 10, or 15-year periods. A 5-year trend can be expected to be dominated by internal variability, with as many positive changes as negative, whereas the 15-year trend should, according to recent literature[5,10,11], reveal differences between the high and low emissions scenarios.

Looking first at unfiltered GSTA trends (hashed histograms in Fig. 5), these expectations are borne out. In 15-year trends (2021–2035), SSP5–8.5 would result in a slight increase relative to the past trend (0.05 °C per decade; see whisker plots), while SSP1–2.6 would, on average, give a 0.1 °C per decade reduction. 5-year trends (2021–2025) show no significant scenario difference and no clear preference towards positive or negative changes. 10-year trends represent a midpoint between these results.

For the filtered results (open histograms in Fig. 5), we find a narrowing of the distribution for all three periods. Even for the 5-year period, we now find more realizations with a reduced rate in SSP1–2.6 relative to 1991–2020, indicating that the filtering method improves separation between high and low emission climate evolution already at this early stage. While the present improvement is still insufficient to allow formal detection of

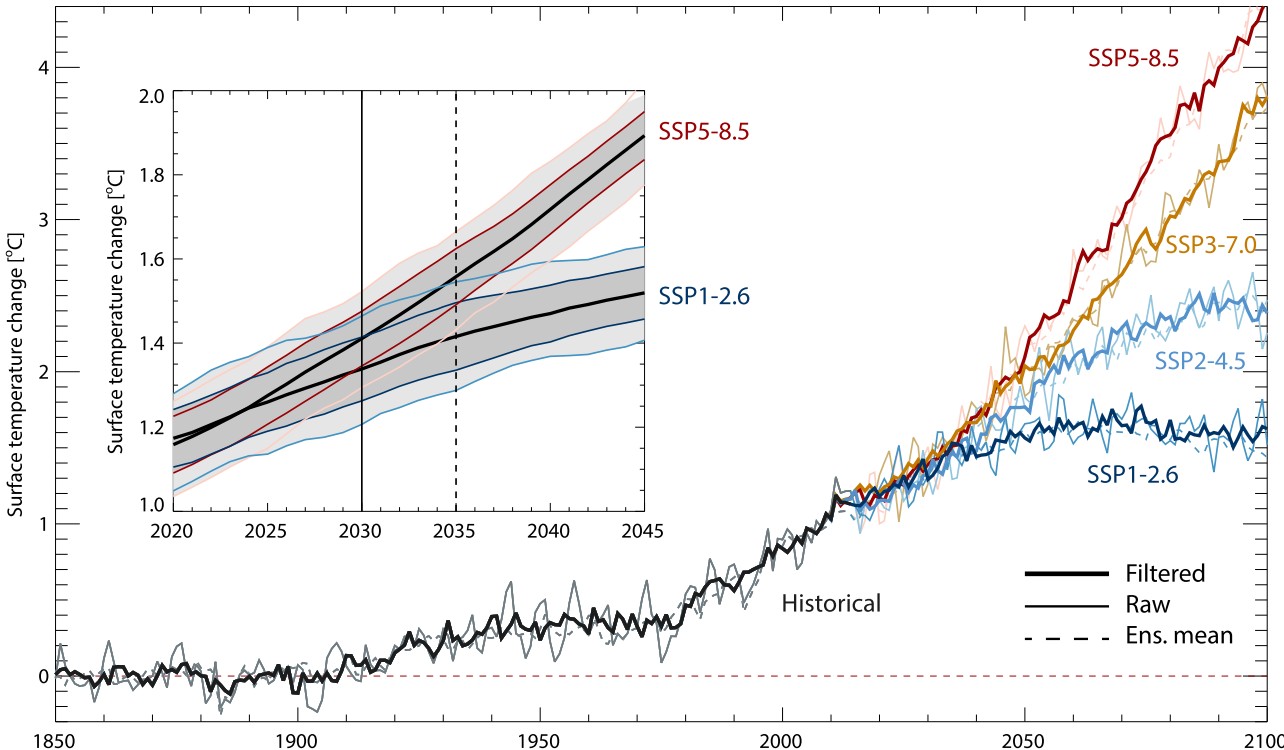

**Fig. 3 Raw and filtered global mean surface temperature anomaly (GSTA) values from one ensemble member of the MPI-ESM1.2 Earth System Model.** Emissions follow historical (black) and Shared Socioeconomic Pathway (SSP) scenario (colored) trajectories. Inset: Mean and ±1 standard deviation ranges for a 10-member initial condition ensemble from MPI-ESM1.2, for raw (light shading) and filtered (dark shading) GSTA values. Vertical lines show the year of emergence in the filtered (solid) and unfiltered (dashed) case, based on 10-year running means.

**Table 1 Earlier emergence of a change in 10-year (left) and 20-year (right) smoothed GSTA in filtered simulations from MPI-ESM1.2, relative to unfiltered simulations.**

|  | 10-year emergence | Mitigation | | 20-year emergence | Mitigation | |
|---|---|---|---|---|---|---|
|  |  | SSP370 | SSP245 | SSP126 |  | SSP370 | SSP245 | SSP126 |
| Baseline | SSP585 | 6 | 5 | 5 | SSP585 | 8 | 6 | 3 |
|  | SSP370 | – | 9 | 6 | SSP370 | – | 7 | 5 |
|  | SSP245 | – | – | 7 | SSP245 | – | – | 5 |

All values are in years and represent the change in the year of emergence derived from comparing a lower emission signal ("Mitigation") to a higher emission situation ("Baseline"). See also Fig. 3 and Methods.

mitigation effects after 5–10 years, it is promising enough that we encourage further investigations into GF approaches that can encompass a broader range of modulations, feedbacks, and processes than captured by the present methodology (e.g., responses to land surface warming patterns, differentiation of the response of surface temperature conditioned on the state of major atmospheric modes of variability, or the influence of other factors such as volcanic eruptions or the amount and geographical distribution of anthropogenic atmospheric aerosols).

**Testing the method using Bayesian calculus.** A final, more rigorous test of whether GF-based filtering really does lead to improved separation between mitigated and unmitigated climate evolutions, is to apply Bayesian calculus of event causation to the question of whether a reduction in warming rate (the "event", in our case) can be said to be "due to" the reduction in emissions. We follow the quantification method laid out in[4], and defined in Methods, on filtered and unfiltered GSTA time series, using both the CMIP6 multi-model ensemble and an extended, emulated 100-member ensemble based on MPI-ESM1.2-LR (see Methods).

The results are shown in Fig. 6. We calculate 5-, 10-, and 15-year warming rates, starting in 2021, and compare them to the preceding 30-year trend (1991–2020). For the single-model emulated ensemble, we find that the filtering induces a marked increase in the probability of causation, whether requiring a necessary (N), sufficient (S), or sufficient and necessary (SN) causation. For the CMIP6 ensemble, we find improvements for 5-year and 15-year rates, whereas the 10-year rates are virtually unaffected. The CMIP6 ensemble displays overall lower probabilities of causation, as could be expected due to the larger variability in ECS and other factors affecting the GSTA response. Overall, this analysis further strengthens the argument that GF-based filtering can lead to an earlier detectability of a GSTA signal from strong emissions mitigation.

**Discussion**

We have shown that (i) GFs can be used to estimate the component of the monthly GSTA that is due to SST variability, including ENSO, (ii) that the filtered GSTA evolution reveals a clearer picture of the short-term warming rate, and (iii) that the

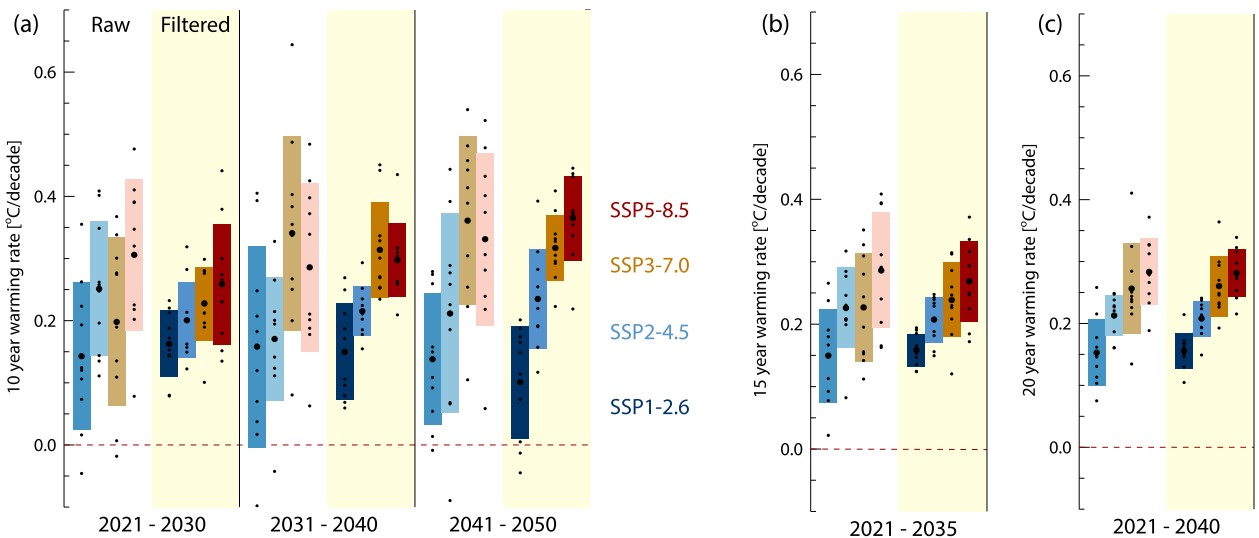

**Fig. 4 Near-term global warming rates for 10 ensemble members from MPI-ESM1.2.** Rates are based on raw (white background, light-colored boxes) and filtered (yellow background, dark-colored boxes) annual mean global mean surface temperature anomaly (GSTA) values. **a** 10-year rates. **b** 15-year rates, and **c** 20-year rates, starting in 2021. Small dots show individual ensemble members, large dots show the 10 ensembles mean. Boxes show ±1 standard deviation.

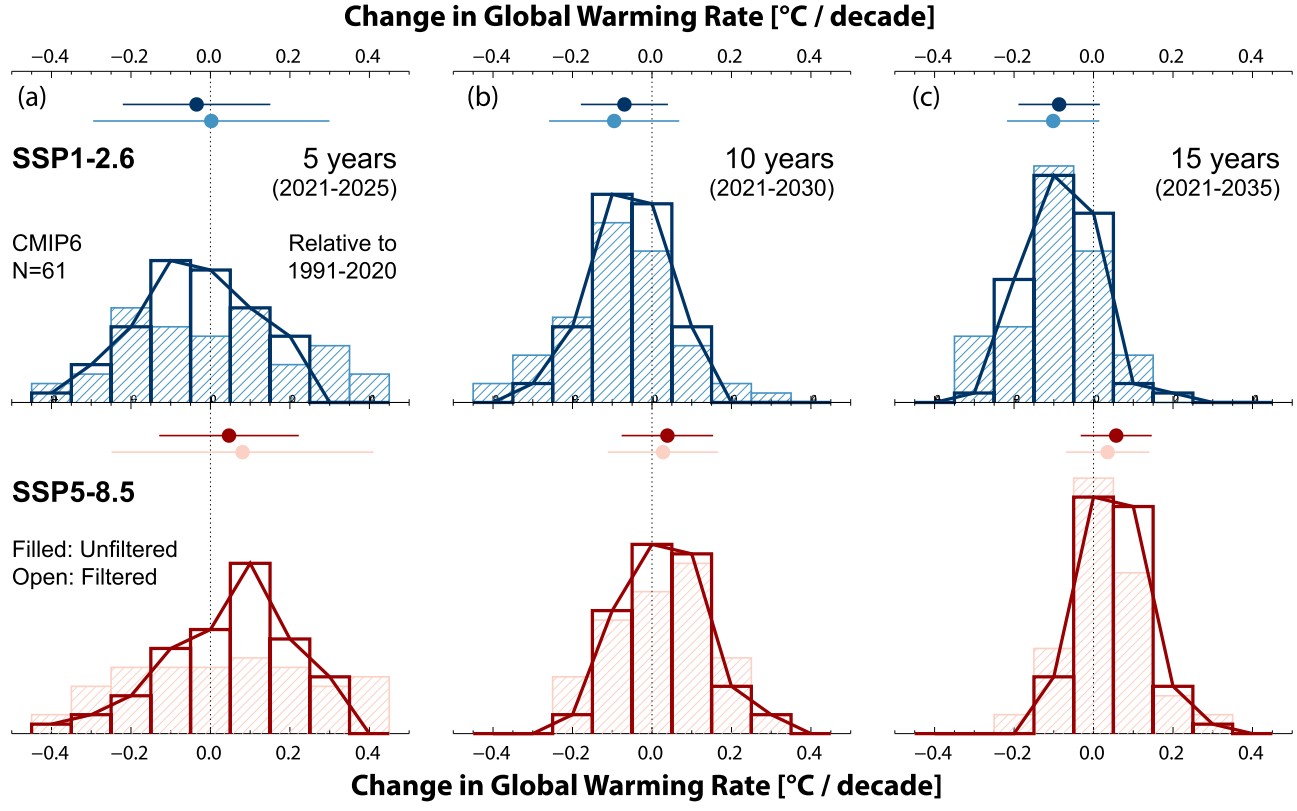

**Fig. 5 Change in global mean warming rate, relative to the recent 30-year period (1991–2020).** Emissions following two future Shared Socioeconomic Pathways (SSP1–2.6 and SSP5–8.5), for a range of CMIP6 models and realizations of internal variability. **a** 5-year rates. **b** 10-year rates. **c** 15-year rates. Hashed histograms show rate changes in the unfiltered simulation results, open histograms and lines show Green's function filtered results. Dots-and-whiskers show the means and ±1 standard deviation ranges of the histograms.

effects of emission mitigation can emerge at an earlier time, relative to a higher emission pathway, if such filtering is applied. A number of caveats should however be mentioned. As used here, the GFs only take into account the modulations to global surface temperature induced by the monthly SST pattern. They will also neglect decadal-scale variability, as well as influences from volcanic eruptions (except where manifested through SST anomaly patterns), anthropogenic aerosols, and mainly atmospheric modes of variability such as the North Atlantic Oscillation. This is however an issue with the current application, and not with the GF method in itself, which could readily be extended to include also these kinds of influences on the GSTA evolution.

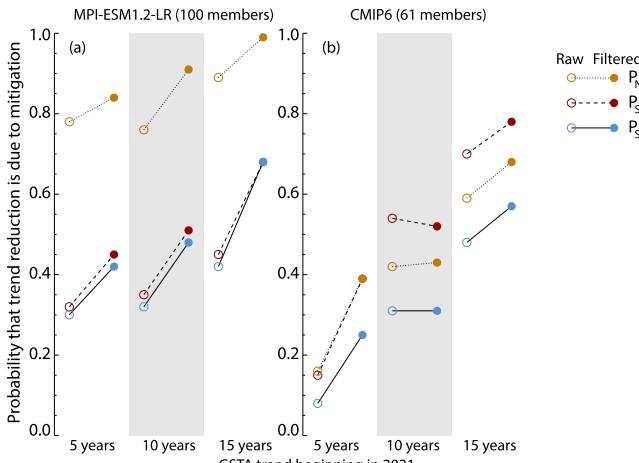

**Fig. 6 Probabilities of causation of a reduction in global mean surface temperature anomaly (GSTA) trend, after a reduction in emissions. a** A 100-member emulated ensemble based on MPI-ESM1.2-LR. **b** CMIP6. Colored symbols show the necessary ($P_N$), sufficient ($P_S$), and sufficient and necessary ($P_{SN}$) probabilities, for raw (open symbols) and filtered (closed symbols) GSTA evolutions.

Modes of atmospheric variability have, e.g., been shown to be sufficient to explain the Eurasian landmass contribution to the 1998–2012 hiatus period[38], indicating that a filtering approach that also takes into account atmospheric modulations unrelated to the SST pattern could be especially powerful. A further question is whether it is reasonable to assume, as we have done here, that GSTA responses can be assumed to occur in the same month as the SST modulation. The GF quantifies the equilibrium response to an SST change, and therefore includes information on the (generally rapid) evolution of the atmosphere after a perturbation. A sensitivity test where the modulation was applied 1–12 months after the initiating SST pattern yielded a progressively lower reduction in interannual variability (see Supplementary Materials.) This indicates that for the presently available GF, the most efficient filtering comes when applying the modulations to the same month as the SST pattern. Finally, the results of the current paper are clearly dependent on the connection between SST patterns and GSTA present in the CESM1 global climate model. Filtering based on GFs derived from other models can be expected to yield qualitatively similar results to the present study, but with differing details for individual years depending on the details of the cloud, moisture, energy transport, and other responses in the host model. A community effort to compare and validate results from filtering with GFs from different models would be a useful next step here.

Achieving the aims of the Paris Agreement and limiting global warming to well below 2 °C requires massive, rapid reductions in anthropogenic emissions of greenhouse gases[39], which in turn is likely to require unprecedented global efforts and public support. Demonstrating that such efforts are having the desired effect will therefore be a crucial task for the scientific community over the coming years and decades. While the connection between greenhouse gas emissions and their atmospheric concentrations can likely be established within years of the implementation of strong mitigation[3,40], most climate impacts—and much of the public discourse—track global surface temperature. Natural, interannual variability, whether it stems from atmospheric processes, insolation, volcanoes, or the SST pattern, will inevitably delay the emergence of a GSTA response to even very strong mitigation. We have shown, however, that even with relatively simple and readily communicable methods, it is possible to

reduce the waiting time until such emergence. In practice, the filtering method could be combined with other approaches to provide an overall probability at the end of a calendar year that the evolution up to and including that year is consistent with known emissions. Regardless of the level of mitigation, this would be a great benefit to the discourse at the science-policy interface, and for communications to the broader public. We, therefore, encourage the further development of filtering approaches such as the GFs used here, and their inclusion—alongside complementary approaches such as the Global Warming Index[13] and others—in the toolkit used by climate scientists when discussing the near-term evolution of anthropogenic climate change, and the emergence of climate responses to stringent emission reductions.

## Methods

**Data sets**. This study makes use of the HadCRUT5 gridded data set of global historical surface temperature anomalies, version 5.0.1.0[8]. Monthly data are available for the period January 1850–December 2020.

ENSO strength is estimated using the ESRL/NOAA Nino3.4, based on HadISST, available for 1870–2020. Data downloaded from https://psl.noaa.gov/gcos_wgsp/Timeseries/Nino34/.

Simulations used are provided for the ScenarioMIP[41] CMIP6 Endorsed MIP[30], and made available to the community through the Earth System Grid Federation (ESGF). We make use of 250-year transient simulations using the CMIP6 historical (1850–2014) and Shared Socioeconomic Pathway (SSP) (2015–2100) emission data sets. Four SSPs are used (SSP1–2.6, SSP2–4.5, SSP3–7.0, SSP5–8.5). See Supplementary Table 1 for a list of models. Only monthly mean temperature data (Global Surface Air Temperature) is used.

In addition, we use 10 ensemble members for each SSP from the MPI Earth System Model (version MPI-ESM1.2-LR[29]). This model was chosen as it has an Equilibrium Climate Sensitivity (ECS = 2.98 K[42]), close to the central value of a recent assessment[32], and because it had provided a comprehensive ensemble simulation for the most recent generation of climate scenarios (the SSPs). The simulations make up an initial condition ensemble, with each transient simulation having the same input emissions but differing in their realized patterns of internal variability.

**Isolating the monthly pattern of natural variability**. For both simulated and observed temperature fields, the pattern of natural, monthly sea surface temperature variability is isolated from the long-term influence of anthropogenic warming via a boxcar smoothing with a 10-year window. For each grid point in the input data set, we construct a time series of monthly temperature anomalies relative to 1850–1900. We calculate a 10-year moving boxcar average, and subtract this mean from the anomalies. By doing this per grid point and per month, we are able to simultaneously take into account global mean temperature increase, the geographical pattern of global warming, and any seasonal differences. Near the endpoints, where there is insufficient data for the 10-year mean, we mirror the data points, which in practice gives extra weight to the last years. Other endpoint treatments were tested, like means with fewer years, but all gave consistently worse results when compared against the multi-ensemble-mean forced response in MPI-ESM-1.2-LR. We also tested the sensitivity to the number of years in the mean, and found improvement in the filtering results up to the 10-year mean but not beyond. The choice of smoothing algorithm (boxcar) was found to have negligible impacts on the results. Note that our chosen method will not remove decadal-scale variability in regional temperature patterns.

**Green's function**. To calculate the modulation of global surface temperature anomaly due to the pattern of sea surface temperatures for each month, we use a pre-calculated GF made with the CESM1.2.1-CAM5.3 Earth System Model. The GF is documented in[25]. Here, we use a monthly resolved, time extended (40 year) version of the calculation[26], to take into account potential differences in GSTA modulation through the year. In all, the GF relates an idealized increase in sea surface temperature at a given location, to resulting influences on radiation, clouds, water vapor, and, ultimately, global mean surface temperature. It provides a means to calculate the modulation of global mean surface temperatures resulting from a given pattern of SST variability. Our specific GF is based on simulations where the SST was individually perturbed in 74 (partially overlapping) ocean patches of 80° longitude and 40° latitude. We use a two-meter surface air temperature to quantify the modulations. A separate GF calculation using skin temperature was also used, as a sensitivity test, which made no quantifiable difference to the results presented here.

**Calculating modulations**. GSTA modulations are calculated by multiplying the GF for that month with the detrended SST pattern from HadCRUT5 or a climate model. See Fig. 1. The total modulation is the sum of the contributions from all ocean-dominated grid points. Note that for this sum to be correctly defined,

calculations must be done in the native resolution of the GF. Hence, both observed and model temperature fields have been regridded to this resolution (2.5° latitude, 1.9° longitude).

**Emergence**. To quantify when a signal from emissions mitigation "emerges" from an assumed non-mitigated background situation, we identify the first year when a 10-year or 20-year running mean of the signal simulations evolves beyond one standard deviation ($\sigma$) around the mean of the background. Specifically, we quantify the means and $\sigma$ of the GSTA values of the MPI-ESM1.2 ensembles, and calculate, e.g., when the 20-year running mean of SSP1–2.6 falls below the $1\sigma$ envelope around the 20-year running mean of SSP5–8.5. This definition is consistent with what was recently used in the IPCC AR6[43].

**Rate calculations**. Rate calculations are performed as simple linear fits to 5-, 10-, 15-, or 20-year series of annual mean GSTA values.

**Estimation of the probability that a trend reduction is due to emissions mitigation**. Following Marotzke[4], we use Boolean algebra of event causation to quantify the probability that a trend reduction after an emission change can be ascribed to that change. By performing this calculation on raw and filtered GSTA data, we can get a quantification of whether the GF-based filtering increases the likelihood of a trend reduction.

If, for a given GSTA time series, we calculate a reduction in the multi-year trend (5, 10, or 15 years in our case), the question is whether the underlying emission change can be said to be a sufficient (S), necessary (N), or sufficient and necessary (SN) condition for the reduction.

Let $P_{SCEN}$ be the probability of a trend reduction in a given scenario, quantified by calculating it for all members of a large initial condition ensemble. Then, using the difference between SSP5–8.5 and SSP1–2.6 as an example, the three probabilities of event causation can be calculated as follows (assuming $P_{SSP1–2.6} > P_{SSP5–8.5}$):

$$P_{NS} = P_{SSP1–2.6} - P_{SSP5–8.5} \tag{1}$$

$$P_{S} = \frac{P_{SSP1–2.6} - P_{SSP5–8.5}}{1 - P_{SSP5–8.5}} \tag{2}$$

$$P_{N} = 1 - \frac{P_{SSP5–8.5}}{P_{SSP1–2.6}} \tag{3}$$

For CMIP6 simulations, we perform the calculation on the full set of 61 realizations described above. For MPI-ESM1.2-LR, where we only have 10 ensemble members per scenario available, we perform a simple emulation to increase the sample size. Following ref. [5], we take the forced GSTA response to be the ensemble mean. We detrend each member by the forced response, and then add independent 50-year periods of internal variability to the forced response to generate a larger number of members that all follow the same overall trend. Here, we calculate 100 such emulated members, and use them for the probability calculations in Fig. 6.

## Data availability

HadCRUT5 historical surface temperature anomalies are available at https://www.metoffice.gov.uk/hadobs/hadcrut5/. All climate model simulations used in this manuscript are publicly available through the ESGF. The GF is documented and made available at https://github.com/mzelinka/greens-function (https://doi.org/10.5281/zenodo.5514146). ENSO strength calculations are available at https://psl.noaa.gov/gcos_wgsp/Timeseries/Nino34/.

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

## Acknowledgements

B.H.S, J.S.F., and M.T.L. acknowledge funding by the Research Council of Norway through the projects CATHY (324182) and QUISARC (248834). The work of M.D.Z. was supported by the U.S. Department of Energy (DOE) Regional and Global Model Analysis program area and was performed under the auspices of the DOE by Lawrence Livermore National Laboratory under Contract DE-AC52-07NA27344. J.M. was supported by the Max Planck Society for the Advancement of Science. The work of CZ was supported by the National Natural Science Foundation of China (Grant No. NSFC 41875095 and 42075127).

## Author contributions

B.H.S. conceived and designed the study, and performed the analysis. C.Z. and M.D.Z. produced and provided the GFs. B.H.S., C.Z., J.S.F., M.T.L., J.M., and M.D.Z. all contributed to shaping the conclusions, discussing and testing the methods, and writing the paper.

## Competing interests

The authors declare no competing interests.
