## [Peer Review File · Nature Communications]

Earlier emergence of a temperature response to mitigation by filtering annual variabilityREVIEWER COMMENTS

Reviewer #1 (Remarks to the Author):

Review of NCOMMS-21-27213-T

This paper proposes to use Green's functions associated with monthly sea surface temperature anomalies, specifically computed from CESM simulations, as a novel approach to the identification of the forced response in global surface temperature anomalies. The authors show their results on observational records (HadCRUT5) and test the validity of the method using an initial condition ensemble of the MPI model, and CMIP6 ScenarioMIP results. In all cases the main finding is that the interannual variations after the filtering is significantly lower, and this helps in calculating decadal trends that appear less variable and more in line with the 30-year trends that are considered a robust identification of the forced signal, cleaned out of the influences of internal variability.

The approach seems rigorous and the results robust. I'm not an expert on the application of Green's functions, but since the method has been well documented and the authors test its results on model simulations and show that their estimates' improvements on unfiltered results appear consistent, I am happy. The authors discuss limitations and ways forward openly and valuably. The ability of the method to shorten the time needed to detect the forced component appears promising.

I do have a couple of questions/suggestions, however.

First, I think there needs to be a clarification of what the authors define "separation", which I have not found anywhere.

Second, since the authors show that their approach "works" on other models than CESM, I think they need to elaborate on what they mean when they say that their approach may be marred by the fact that their Green's functions are estimated from CESM. What would a test of this dependence look like? What type of results would be expected if another model was used to estimate them?

Third, since the authors highlight the value of their results by repeatedly showing that they are in alignment with a 30 year trend, I wonder if a necessary test of the gains of this approach on a simple 30-year unfiltered trend computation, rather than a 10-year unfiltered trend, would make better sense. What are the gains of this approach over considering the longer span (ending at each year that is now the last year of the 10-year span used in the filtered trend computation)? To be clear, I'm not suggesting that the authors compare two 30-year trends (filtered or unfiltered) but that they compare their 10-year filtered trends to values of the trends computed over the 30-year period ending at the same time. It would seem to me that if the goal is to inform policy makers about the efficacy of mitigation, possibly the longer unfiltered trend could be as effective...but I'm happy to be proved wrong.

Last, I would like to call the authors' attention upon some work (mainly but not only by Clara Deser and colleagues in the last 5 or 6 years). I'm going to list a few papers, which would seem to me worth considering in an otherwise nicely surveyed background literature. I realize some of this work was developed for future projections (then later extended to observations) and its application has been mostly to regional estimates of time of emergence, but it would seem to me germane enough to the discussion to be worth citing.

Deser, C., L. Terray, and A. S. Phillips, 2016: Forced and internal components of winter air temperature trends over North America during the past 50 years: Mechanisms and implications. *J. Climate*, 29, 2237–2258, doi:10.1175/JCLI-D-15-0304.1.

Lehner, F., Deser, C., & Terray, L. (2017). Toward a New Estimate of "Time of Emergence" of Anthropogenic Warming: Insights from Dynamical Adjustment and a Large Initial-Condition Model Ensemble, *Journal of Climate*, 30(19), 7739-7756. Retrieved Aug 4, 2021, from <https://journals.ametsoc.org/view/journals/clim/30/19/jcli-d-16-0792.1.xml>

Wills, R. C. J., Battisti, D. S., Armour, K. C., Schneider, T., & Deser, C. (2020). Pattern Recognition Methods to Separate Forced Responses from Internal Variability in Climate Model Ensembles and Observations, *Journal of Climate*, 33(20), 8693-8719. Retrieved Aug 4, 2021, from <https://journals.ametsoc.org/view/journals/clim/33/20/jcliD190855.xml>

Thompson, D. W. J., Barnes, E. A., Deser, C., Foust, W. E., & Phillips, A. S. (2015). Quantifying the Role of Internal Climate Variability in Future Climate Trends, *Journal of Climate*, 28(16), 6443-6456. Retrieved Aug 4, 2021, from <https://journals.ametsoc.org/view/journals/clim/28/16/jcli-d-14-00830.1.xml>

Sippel, S., Meinshausen, N., Merrifield, A., Lehner, F., Pendergrass, A. G., Fischer, E., & Knutti, R. (2019). Uncovering the Forced Climate Response from a Single Ensemble Member Using Statistical Learning, *Journal of Climate*, 32(17), 5677-5699. Retrieved Aug 4, 2021, from <https://journals.ametsoc.org/view/journals/clim/32/17/jcli-d-18-0882.1.xml>

Reviewer #2 (Remarks to the Author):

This paper demonstrates a new method to remove internal variability from global-mean surface temperature trends, using a combination of a simple statistical method to identify short-term SST anomalies and a model-based Green's function linking these SST anomalies to global land-surface temperature anomalies. Furthermore, it demonstrates how this can improve detection of the impacts on GSTA of emissions reductions. Both of these analyses represent valuable scientific contributions; however, more could be done to compare this method to existing methods, and the framing of this paper is problematic in a number of places. These concerns revolve primarily around the author's repeated claims that this method can do something that existing methods cannot, when in fact any number of existing statistical methods could achieve a similar result. This can be addressed either (1) by reframing the paper in such a way that this method is presented as a novel approach that complements existing approaches, without necessarily being superior to them, or (2) actually comparing to existing methods and demonstrating that this method is superior to those approaches. Specific parts of the paper where this needs to be addressed are identified below. I think this paper will make a valuable contribution to the literature, and it would likely be suitable for publication after a round of major revisions to address these concerns.

Major comments:

1. You need to work on your argument for why this method is preferable to pattern-based statistical methods (e.g., lines 54-58 and 79-80). It is not clear what you mean by "lack treatment of direct physical connections between modes of variability, anthropogenic forcing and their net influence in a given year or period." If you are saying that (for example) ENSO variability rectifies in such a way that it influences the rate of anthropogenic warming, then I think there is tenuous evidence for this, if any. Purely statistical methods certainly require some bold assumptions (e.g., being able to identify modes of variability based on their similarity with variability in models), but your explanation here doesn't seem to get at the heart of the issue, and your reliance on model-based Green's function is a similarly large assumption. I think it would be preferable to frame this in a way where all of these approaches are complementary, rather than trying to make an argument that your approach is superior, which is a much more difficult argument to make convincingly (you haven't yet).

2. Related to the previous comment, it is not obvious that your approach is much different from the statistical methods you are claiming to improve upon. You are using a fairly ad-hoc method that identifies SST anomalies that deviate from a 10-year running-mean trend, then running that through a Green's function for global land-surface temperature. Something similar could also be accomplished by computing the EOFs for the deviation from the 10-year running mean trend, then computing the regression of global land-surface temperature against the leading EOFs. This would accomplish a similar result in a purely statistical way and would use observation-based regressions instead of model-based Green's functions. Your approach is an interesting alternative to try, but it is not clearly

preferable, and you don't compare it with any existing methods. You could address this either by comparing to an existing method (even something as simple removing the regression against the Niño3.4 index) or by toning down your language about this method being unique and superior to existing methods.

One statistical detail of your methodology, namely the 10-year running mean trend removal, could have an outsized impact on your results, and the sensitivity to the length of this running mean should be tested (i.e., please elaborate on your statement on what exactly you tested in your statement on line 247-248).

3. Overall, I appreciate your discussion of how this method (or others) could be used to detect the impact of emissions reductions. However, it would be helpful to elaborate somewhere in a few sentences how you would envision this in practice, especially considering that there is only one realization of the real world, and there will be no corresponding high-emissions scenario to compare to, other than in models, where the climate sensitivity could be different than the real world.

4. Throughout the text (or at least on line 24), please distinguish whether you are talking about surface skin temperature or surface air temperature.

Line comments:

- Line 24: Do you mean surface air temperature?
- Line 28 & 30: I suggest replacing "rates" with "warming rates"
- Line 53-54: A few more recent papers use statistical techniques to separate anthropogenic warming from other influences and probably deserve mentioning considering their focus on similar questions to your paper (Frankignoul et al. 2017; Sippel et al. 2019; Wills et al. 2020)
- Line 54-58: There is a bit too much in one sentence here and it's hard to follow. Considering the importance of this sentence in differentiating your study from these past studies, please consider expanding this into multiple sentences. See also Comment 1.
- Line 59: missing 'is' after that
- Line 61-62: Please cite Dong et al. (2019) here as well.
- Figure 1: It is initially confusing that the Green's functions are different in the different panels, because you don't explicitly say in the main text that the Green's function varies seasonally. Please clarify this detail in the text.
- Line 63-65: This sentence is not very clear. I'm assuming by residual GTSA you mean after removing that associated with the SST pattern, but if so, then you should say so. In that case, you should be more specific than "decadal variability", because decadal variability mostly manifests in SST patterns. You might also want to be more specific about which aspect of the underlying anthropogenic warming is relevant here.
- Line 96: This statement is wrong. The aforementioned statistical papers (Frankignoul et al. 2017; Sippel et al. 2019; Wills et al. 2020) all do exactly this, and they are certainly not the only ones.
- Line 126: Most readers will know that MPI-ESM1.1 has a 100-member ensemble for each of 3 different RCP scenarios, so it might be worth mentioning in the text why you only used the 10-member MPI-ESM1.2 ensemble. I'm not suggesting that you need to use the larger (and older) ensemble, it's just worth mentioning. CanESM5 also has a 50-member ensemble for each of the 4 SSP scenarios you look at.
- Line 158: Not sure what you mean by 'consistent' here. Consistent in what way?
- Figure 5: A reader quickly glancing at this figure might be confused by the use of the change in global warming rate, rather than the absolute global warming rate. This is perfectly fine quantity to use, but it might help with the reader's comprehension if you put one larger and bolded "Change in Global Warming Rate" label on the top and bottom of the figure, rather than 3 redundant labels.
- Line 177: What do you mean by "encompass a broader range of modulations, feedbacks and processes". Aren't these Green's functions capturing all impacts of SST patterns on global land temperature? Do you mean remote impacts on SST and sea-ice surface temperature as well (necessitating a different model setup)? Please elaborate.
- Line 183: extraneous 'and'
- Line 188: "latter set of modes" is unclear. How about "modes of atmospheric variability"?

• Line 200: This is the first time CESM1 is mentioned in the main text. Please mention which model the Green's functions are based on earlier (if you want to mention it in the main text at all).

References:

Dong, Y., C. Proistosescu, K. C. Armour, and D. S. Battisti, 2019: Attributing historical and future evolution of radiative feedbacks to regional warming patterns using a Green's function approach: The preeminence of the Western Pacific. *J. Climate*, 32, 5471–5491, <https://doi.org/10.1175/JCLI-D-18-0843.1>.

Frankignoul, C., G. Gastineau, and Y.-O. Kwon, 2017: Estimation of the SST response to anthropogenic and external forcing and its impact on the Atlantic multidecadal oscillation and the Pacific decadal oscillation. *J. Climate*, 30, 9871–9895, <https://doi.org/10.1175/JCLI-D-17-0009.1>.

Sippel, S., N. Meinshausen, A. Merrifield, F. Lehner, A. G. Pendergrass, E. Fischer, and R. Knutti, 2019: Uncovering the forced climate response from a single ensemble member using statistical learning. *J. Climate*, 32, 5677–5699, <https://doi.org/10.1175/JCLI-D-18-0882.1>.

Wills, R. C. J., D. S. Battisti, K. C. Armour, T. Schneider, T., and C. Deser, 2020: Pattern Recognition Methods to Separate Forced Responses from Internal Variability in Climate Model Ensembles and Observations. *J. Climate*, 33, 8693–8719, <https://doi.org/10.1175/JCLI-D-19-0855.1>.

Earlier emergence of a temperature response to mitigation by filtering annual variability

Samset *et al.*

Responses to reviewer comments

We thank both reviewers for their very constructive and insightful comments. Overall, we have taken the suggestions into account, and hope that you will find the revised manuscript strengthened and of continued interest. We give point-by-point responses (in red) below, but broadly, our substantive revisions consist of:

- 1) A reframing of the introduction and discussion, as suggested by both reviewers, to take into account a broader set of existing studies and methods
- 2) The addition of two new methods for quantifying the improvements brought by the currently available Green's function: An quantitative emergence analysis (in Figure 3 and the relevant discussion) that replaces the earlier argument based on separation, and a framework of Bayesian event causation drawing on (Marotzke, 2019) (new Figure 6)

Beyond this, the revisions are important for clarity, and for putting the work in a broader context, but do not otherwise alter the overall conclusions of the original manuscript.

Reviewer #1 (Remarks to the Author):

Review of NCOMMS-21-27213-T

This paper proposes to use Green's functions associated with monthly sea surface temperature anomalies, specifically computed from CESM simulations, as a novel approach to the identification of the forced response in global surface temperature anomalies. The authors show their results on observational records (HadCRUT5) and test the validity of the method using an initial condition ensemble of the MPI model, and CMIP6 ScenarioMIP results. In all cases the main finding is that the interannual variations after the filtering is significantly lower, and this helps in calculating decadal trends that appear less variable and more in line with the 30-year trends that are considered a robust identification of the forced signal, cleaned out of the influences of internal variability.

The approach seems rigorous and the results robust. I'm not an expert on the application of Green's functions, but since the method has been well documented and the authors test its results on model simulations and show that their estimates' improvements on unfiltered results appear consistent, I am happy. The authors discuss limitations and ways forward openly and valuably. The ability of the method to shorten the time needed to detect the forced component appears promising.

I do have a couple of questions/suggestions, however.

First, I think there needs to be a clarification of what the authors define "separation", which I have not found anywhere.

Thanks. In the revision, we have exchanged «separation» for the term «emergence», and introduced the definition used throughout the recent IPCC AR6 WG1 report:

«To quantify when a signal from emissions mitigation "emerges" from an assumed non-mitigated background situation, we identify the first year when a 20-year running mean of the signal simulations evolves beyond one standard deviation (σ) around the mean of the background.»

(from the revised Methods). Further, we have added a quantification of emergence according to this definition, visually in Figure 2 for one combination of signal and background, and quantitatively in Table 1 which now shows the number of years by which emergence moves earlier in the filtered situation.

Second, since the authors show that their approach “works” on other models than CESM, I think they need to elaborate on what they mean when they say that their approach may be marred by the fact that their Green’s functions are estimated from CESM. What would a test of this dependence look like? What type of results would be expected if another model was used to estimate them?

What we refer to here is simply the fact that the connection between a given SST pattern and a global surface warming will necessarily depend on the responses in the host model of the Green’s function. We have extended the discussion as follows:

«Finally, the results of the current paper are clearly dependent on the connection between SST patterns and GSTA present in the CESM1 global climate model. Filtering based on with GFs derived from other models can be expected to yield qualitatively similar results to the present study, but with differing details for individual years depending on the details of the cloud, moisture, energy transport and other responses in the host model. A community to effort to compare and validate results from filtering with GFs from different would be a useful next step here. “

Third, since the authors highlight the value of their results by repeatedly showing that they are in alignment with a 30 year trend, I wonder if a necessary test of the gains of this approach on a simple 30-year unfiltered trend computation, rather than a 10-year unfiltered trend, would make better sense. What are the gains of this approach over considering the longer span (ending at each year that is now the last year of the 10-year span used in the filtered trend computation)? To be clear, I’m not suggesting that the authors compare two 30-year trends (filtered or unfiltered) but that they compare their 10-year filtered trends to values of the trends computed over the 30-year period ending at the same time. It would seem to me that if the goal is to inform policy makers about the efficacy of mitigation, possibly the longer unfiltered trend could be as effective...but I’m happy to be proved wrong.

Thanks for the suggestion. This makes for a slightly different test, so we included it in addition to the 10-year running mean test we already had in Figure 2. We include the improvement in residual standard deviation of the 10-year means (backwards looking) when compared against a 30-year mean ending in the same year. The results show a clear improvement from the filtering, although not as substantial as for the GSTA values themselves. The one disadvantage of this method is that 30-year means will not capture more rapid changes like the period 1940-1960 (as Figure 2 clearly shows), but for the present issue – which is isolating deviations from a relatively uniform trend lasting more than 30 years – the approach works well.

Last, I would like to call the authors’ attention upon some work (mainly but not only by Clara Deser and colleagues in the last 5 or 6 years). I’m going to list a few papers, which would seem to me worth considering in an otherwise nicely surveyed background literature. I realize some of this work was developed for future projections (then later extended to observations) and its application has been mostly to regional estimates of time of emergence, but it would seem to me germane enough to the

discussion to be worth citing.

Thanks, we agree that these studies warrant mentions and inclusion. The introduction and discussion have been revised accordingly, and a number of additional citations added.

Reviewer #2 (Remarks to the Author):

This paper demonstrates a new method to remove internal variability from global-mean surface temperature trends, using a combination of a simple statistical method to identify short-term SST anomalies and a model-based Green's function linking these SST anomalies to global land-surface temperature anomalies. Furthermore, it demonstrates how this can improve detection of the impacts on GSTA of emissions reductions. Both of these analyses represent valuable scientific contributions; however, more could be done to compare this method to existing methods, and the framing of this paper is problematic in a number of places. These concerns revolve primarily around the author's repeated claims that this method can do something that existing methods cannot, when in fact any number of existing statistical methods could achieve a similar result. This can be addressed either (1) by reframing the paper in such a way that this method is presented as a novel approach that complements existing approaches, without necessarily being superior to them, or (2) actually comparing to existing methods and demonstrating that this method is superior to those approaches. Specific parts of the paper where this needs to be addressed are identified below. I think this paper will make a valuable contribution to the literature, and it would likely be suitable for publication after a round of major revisions to address these concerns.

We thank the reviewer for these very constructive and thoughtful comments. We've addressed them individually below, but broadly, we agree with the sentiment expressed here that the Green's function method should be more clearly presented as complementary to the existing literature. An intercomparison of methods would be a next step, likely requiring a larger author team. For details of our edits, and further responses to the comments, see below.

Major comments:

1. You need to work on your argument for why this method is preferable to pattern-based statistical methods (e.g., lines 54-58 and 79-80). It is not clear what you mean by "lack treatment of direct physical connections between modes of variability, anthropogenic forcing and their net influence in a given year or period." If you are saying that (for example) ENSO variability rectifies in such a way that it influences the rate of anthropogenic warming, then I think there is tenuous evidence for this, if any. Purely statistical methods certainly require some bold assumptions (e.g., being able to identify modes of variability based on their similarity with variability in models), but your explanation here doesn't seem to get at the heart of the issue, and your reliance on model-based Green's function is a similarly large assumption. I think it would be preferable to frame this in a way where all of these approaches are complementary, rather than trying to make an argument that your approach is superior, which is a much more difficult argument to make convincingly (you haven't yet).

Thanks. We agree that these passages should be reframed, to clarify that our approach is complementary to the existing literature. We do believe that the traceability that is possible through Green's functions is an advantage, but it's true that this has not been fully explored in the present study (it is implicitly discussed in the references we cite though, notably Zhou et al. 2020 and 2017) and also that this doesn't make our approach superior in general. We've rephrased the relevant passages, which now read:

«These approaches all have strengths and weaknesses. Some rely primarily on emission inventories and model estimates of the links between emissions, radiative forcing and surface temperature responses. Others lack direct treatment of the physical connections between modes of variability, anthropogenic forcing and their net influence in a given year or period, relying instead on the assumption that we can identify modes of variability in a model based on similarities with real-world observations.

In the following, we present an approach to filtering interannual variability that is complementary to existing techniques, based around recently developed Green's Functions...»

and

«This makes it complementary to the forcing-based approach mentioned above, and also to statistical approaches that do not retain information on physical processes linking modes of variability to temperature responses.»

2. Related to the previous comment, it is not obvious that your approach is much different from the statistical methods you are claiming to improve upon. You are using a fairly ad-hoc method that identifies SST anomalies that deviate from a 10-year running-mean trend, then running that through a Green's function for global land-surface temperature. Something similar could also be accomplished by computing the EOFs for the deviation from the 10-year running mean trend, then computing the regression of global land-surface temperature against the leading EOFs. This would accomplish a similar result in a purely statistical way and would use observation-based regressions instead of model-based Green's functions. Your approach is an interesting alternative to try, but it is not clearly preferable, and you don't compare it with any existing methods. You could address this either by comparing to an existing method (even something as simple removing the regression against the Niño3.4 index) or by toning down your language about this method being unique and superior to existing methods.

As above, we agree on the need for a different language, and have rephrased as described under comment 1 above. Regarding the regression technique, this is an excellent suggestion – and indeed something that we are already working on independently of the present study – but we feel it would require too much additional material to add it here. An intercomparison of available techniques is indeed a natural next step (as is the possibility of combining complementary approaches), but would require a different kind of analysis and presentation.

One statistical detail of your methodology, namely the 10-year running mean trend removal, could have an outsized impact on your results, and the sensitivity to the length of this running mean should be tested (i.e., please elaborate on your statement on what exactly you tested in your statement on line 247-248).

This was tested at an early stage of the analysis, as part of our sensitivity testing, but we agree that the discussion was too brief. It now reads:

«Near the endpoints, where there is insufficient data for the 10-year mean, we mirror the data points, which in practice gives extra weight to the last years. Other endpoint treatments were tested, like means with fewer years, but all gave consistently worse results when compared against the multi-ensemble-mean forced response in MPI-ESM-1.2-LR. We also tested the sensitivity to the number of years in the mean, and found improvement in the filtering results up to the 10-year mean but not beyond. The choice of smoothing algorithm (boxcar) was found to have negligible impacts on the results. Note that our chosen method will not remove decadal scale variability in regional temperature patterns.»

3. Overall, I appreciate your discussion of how this method (or others) could be used to detect the impact of emissions reductions. However, it would be helpful to elaborate somewhere in a few sentences how you would envision this in practice, especially considering that there is only one realization of the real world, and there will be no corresponding high-emissions scenario to compare to, other than in models, where the climate sensitivity could be different than the real world.

While such usage would require development of tools beyond what we present in the current manuscript, we do have some ideas on how this could be achieved. We have briefly outlined them in the revised conclusion section:

“In practice, the filtering method could be combined with other approaches to provide an overall probability at the end of a calendar year, that the evolution up to and including that year is consistent with known emissions. Regardless of the level of mitigation, this would be a great benefit to the discourse at the science-policy interface, and for communications to the broader public. We therefore encourage the further development of filtering approaches such as the Green’s functions used here, and their inclusion – alongside complementary approaches such as the Global Warming Index (Haustein et al., 2017) and others – in the toolkit used by climate scientists when discussing the near-term evolution of anthropogenic climate change, and the emergence of climate responses to stringent emission reductions.”

4. Throughout the text (or at least on line 24), please distinguish whether you are talking about surface skin temperature or surface air temperature.

Generally we refer to surface air temperature here; this has been specified. The analysis is not sensitive to the choice of air or skin temperature as the value used in the Green’s function (we have used both, as a sensitivity test), though in the main analysis we use air temperature to be most broadly consistent with the model simulations.

Line comments:

- Line 24: Do you mean surface air temperature?

We’re deliberately not distinguishing between surface temperature and surface air temperature at this point. There’s a technical difference, and differing evolution of GMST and GSAT in the models, but our main argument about the visibility of «global surface temperature» and its variability is independent of this distinction. Later in the manuscript, where we use one type or the other, we do make the distinction though.

- Line 28 & 30: I suggest replacing “rates” with “warming rates”

Done. Thanks.

- Line 53-54: A few more recent papers use statistical techniques to separate anthropogenic warming from other influences and probably deserve mentioning considering their focus on similar questions to your paper (Frankignoul et al. 2017; Sippel et al. 2019; Wills et al. 2020)

Done. Thanks for the references.

- Line 54-58: There is a bit too much in one sentence here and it’s hard to follow. Considering the importance of this sentence in differentiating your study from these past studies, please consider expanding this into multiple sentences. See also Comment 1.

Thanks. The sentence has been revised and expanded; see our response to Comment 1 above.

- Line 59: missing 'is' after that

Done. Thanks.

- Line 61-62: Please cite Dong et al. (2019) here as well.

Done. Our intention here was to primarily cite the GF literature that we actually use, but we see that it can be taken as a generic statement so we appreciate the suggestion.

- Figure 1: It is initially confusing that the Green's functions are different in the different panels, because you don't explicitly say in the main text that the Green's function varies seasonally. Please clarify this detail in the text.

This is already stated in several places («Briefly, given a pattern of sea surface temperature, the GF provides an estimate of the influence of that pattern on the global land surface temperature anomaly for the given month or year.»; «The resulting GSTA pattern is multiplied with the GF for the corresponding month.»), but the comment indicates that we need additional clarity. We've expanded a key sentence like this: «and that the strength of this modulation depends on the pattern of SST anomalies and the month in which it occurs.»

- Line 63-65: This sentence is not very clear. I'm assuming by residual GTSA you mean after removing that associated with the SST pattern, but if so, then you should say so. In that case, you should be more specific than «decadal variability», because decadal variability mostly manifests in SST patterns. You might also want to be more specific about which aspect of the underlying anthropogenic warming is relevant here.

Thanks. We've expanded the sentence accordingly:

«Our aim is to quantify, as far as possible, the contribution to GSTA for a given year that can be related directly to responses from the realized SST pattern. The remaining surface temperature anomaly will then be a combination of internal variability on decadal and longer timescales (which is still present after our removal of the 10-year trend), other feedbacks and modulations (such as responses to warming patterns over land), and the underlying influence of anthropogenic global warming (notably patterns such as Arctic amplification).»

- Line 96: This statement is wrong. The aforementioned statistical papers (Frankignoul et al. 2017; Sippel et al. 2019; Wills et al. 2020) all do exactly this, and they are certainly not the only ones.

Thanks. We've removed the statement.

- Line 126: Most readers will know that MPI-ESM1.1 has a 100-member ensemble for each of 3 different RCP scenarios, so it might be worth mentioning in the text why you only used the 10-member MPI-ESM1.2 ensemble. I'm not suggesting that you need to use the larger (and older) ensemble, it's just worth mentioning. CanESM5 also has a 50-member ensemble for each of the 4 SSP scenarios you look at.

Thanks. We're using the other ensembles for other analyses, but here we wanted (i) to stick to a CMIP6 model version that had the SSPs rather than the RCPs (mainly to get at the more realistic aerosol pathways, though that isn't a focus of the present paper), and (ii) to have a model that is in the middle of the ECS range. CanESM5 has such high warming rates that it might give non-

representative results from the filtering. We've added a brief additional comment on this to the Methods section.

- Line 158: Not sure what you mean by 'consistent' here. Consistent in what way?

We only meant that the results come from the same models (and number of ensembles) for all scenarios – i.e. that they are directly comparable. This has been clarified.

- Figure 5: A reader quickly glancing at this figure might be confused by the use of the change in global warming rate, rather than the absolute global warming rate. This is perfectly fine quantity to use, but it might help with the reader's comprehension if you put one larger and bolded "Change in Global Warming Rate" label on the top and bottom of the figure, rather than 3 redundant labels.

Thanks. This has been added.

- Line 177: What do you mean by "encompass a broader range of modulations, feedbacks and processes". Aren't these Green's functions capturing all impacts of SST patterns on global land temperature? Do you mean remote impacts on SST and sea-ice surface temperature as well (necessitating a different model setup)? Please elaborate.

Thanks. The following explanation has been added:

"(e.g. responses to land surface warming patterns, differing responses under atmospheric modes of variability, or the influence of other factors such as volcanic eruptions or the amount and geographical distribution of anthropogenic atmospheric aerosols)."

- Line 183: extraneous 'and'

Thanks. Fixed.

- Line 188: "latter set of modes" is unclear. How about "modes of atmospheric variability"?

Thanks. Fixed.

- Line 200: This is the first time CESM1 is mentioned in the main text. Please mention which model the Green's functions are based on earlier (if you want to mention it in the main text at all).

Thanks, good catch. Which model is used is important information, so we've now mentioned it both in the main text and in the caption for Figure 1 (in addition to Methods).

Haustein, K., Allen, M. R., Forster, P. M., Otto, F. E. L., Mitchell, D. M., Matthews, H. D., & Frame, D. J. (2017). A real-time Global Warming Index. *Sci Rep*, 7(1), 15417. doi:10.1038/s41598-017-14828-5

Marotzke, J. (2019). Quantifying the irreducible uncertainty in near-term climate projections. *WIREs Climate Change*, 10(1). doi:10.1002/wcc.563

REVIEWER COMMENTS

Reviewer #1 (Remarks to the Author):

I am fully satisfied with the revised version of this work and I am happy to recommend its publication in Nature Communications.

Reviewer #2 (Remarks to the Author):

The authors have done a good job with the revisions, and the paper is interesting, clear, and well written. This paper will make a valuable contribution to the literature on separating forced and unforced components of climate change and detecting the emergence of emissions reductions. I have a few additional minor suggestions, but I otherwise recommend publication.

Line 57-61: This paragraph now better acknowledges the diversity of methods and that they all have advantages and disadvantages. However, none of the references I suggested in the previous round (Frankignoul et al. 2017; Sippel et al. 2019; Wills et al. 2020) rely on “the assumption that we can identify model-derived modes of variability also in real-world observations.” All of these papers present a method that can identify patterns from observations alone, then remove them from the observations, without using models. They do not include a direct treatment of the physical connection between SST patterns and land-temperature responses, so it would make more sense to focus on that aspect here instead.

Figure 1: Green bars should be labeled interannual variability I would think, or simply variability if you want to save space.

Figure 1: Please also add some more detailed description of what the Green's function shows, either in the text, in the caption, or in the methods (or all three). For example, ‘the Green's function shows the contribution of SST warming in a given grid cell to the GSTA anomaly’. This will be a confusing plot for readers who are not familiar with GFs.

Figure 2: Please consider increasing the height of the middle (green) panel. Note, panels are not labeled, even though the caption refers to panel (a).

Figure 2c: I would expect the variance about the 30-year rates to be lower if you used the same center point, not the same end point. This is mostly a problem for the inset, which I'm not sure is needed anyway.

Line 148: Repeated word ‘mean’ in “10-year running mean over the ensemble mean” doesn't completely make sense to me

Line 151: At end of sentence, it would be air readability if you added “, in 2030 instead of 2035.”

Line 152: At end of sentence, it would be helpful to say “resulting from differences in emissions pathways”

Line 180: by ‘results’, do you mean ‘simulations’?

Figure 6: “Colors” -> “Colored symbols”, otherwise you are implying that the color has quantitative information in it (i.e., necessitating a colorbar).

Minor thing, but in Figure 6 and its description in the text, I would find it more intuitive if you listed the probabilities in the order of how strong a criterion they are (i.e., P_N, P_S, P_NS)

Line 233-234: repeated word 'modulations'

Line 241: extra word - "on with"

Line 245: missing word – "models"

Responses to reviewer comments

Reviewer #1 (Remarks to the Author):

I am fully satisfied with the revised version of this work and I am happy to recommend its publication in Nature Communications.

Thanks!

Reviewer #2 (Remarks to the Author):

The authors have done a good job with the revisions, and the paper is interesting, clear, and well written. This paper will make a valuable contribution to the literature on separating forced and unforced components of climate change and detecting the emergence of emissions reductions. I have a few additional minor suggestions, but I otherwise recommend publication.

We greatly appreciate the detailed and useful responses made by the reviewer.

Line 57-61: This paragraph now better acknowledges the diversity of methods and that they all have advantages and disadvantages. However, none of the references I suggested in the previous round (Frankignoul et al. 2017; Sippel et al. 2019; Wills et al. 2020) rely on “the assumption that we can identify model-derived modes of variability also in real-world observations.” All of these papers present a method that can identify patterns from observations alone, then remove them from the observations, without using models. They do not include a direct treatment of the physical connection between SST patterns and land-temperature responses, so it would make more sense to focus on that aspect here instead.

Good point. The text has been revised accordingly:

These approaches all have strengths and weaknesses. Some rely primarily on emission inventories and model estimates of the links between emissions, radiative forcing and surface temperature responses, and on the assumption that we can identify model-derived modes of variability also in real-world observations. Others identify and subtract modes of variability based on observations alone, but do not include a direct treatment of the physical connection between SST patterns and land temperature responses.

Figure 1: Green bars should be labeled interannual variability I would think, or simply variability if you want to save space.

Done

Figure 1: Please also add some more detailed description of what the Green’s function shows, either in the text, in the caption, or in the methods (or all three). For example, ‘the Green’s function shows the contribution of SST warming in a given grid cell to the GSTA anomaly’. This will be a confusing plot for readers who are not familiar with GFs.

Done.

Figure 2: Please consider increasing the height of the middle (green) panel. Note, panels are not labeled, even though the caption refers to panel (a).

The panel references have been updated in accordance with the journal policy. We've tried various sizes for the middle panel, but feel that if the size of the figure is to be kept manageable then it is difficult to resize without losing the key messages (which are in the top and bottom panels). Hence, we hope we can retain the current layout, which has been through some "user testing" in the sense that it's been shown and discussed with a number of colleagues.

Figure 2c: I would expect the variance about the 30-year rates to be lower if you used the same center point, not the same end point. This is mostly a problem for the inset, which I'm not sure is needed anyway.

The RSD result is in fact quite insensitive to this particular choice. The inset came about in response to another comment, so we'd prefer to leave it in.

Line 148: Repeated word 'mean' in "10-year running mean over the ensemble mean" doesn't completely make sense to me

Agree. We're trying to convey that we first take the mean of the 10 ensemble members, and then find a 10-year running mean from the resulting time series. We've rephrased as follows:

"Defining emergence as the year in which a 10-year running mean over the ensemble average response in the mitigated situation moves outside the plume for the unmitigated situation..."

Line 151: At end of sentence, it would be air readability if you added ", in 2030 instead of 2035."

Done.

Line 152: At end of sentence, it would be helpful to say "resulting from differences in emissions pathways"

Done.

Line 180: by 'results', do you mean 'simulations'?

Yes. Changed.

Figure 6: "Colors" -> "Colored symbols", otherwise you are implying that the color has quantitative information in it (i.e., necessitating a colorbar).

Done.

Minor thing, but in Figure 6 and its description in the text, I would find it more intuitive if you listed the probabilities in the order of how strong a criterion they are (i.e., P_N, P_S, P_NS)

Agree, fixed.

Line 233-234: repeated word 'modulations'

Rephrased slightly.

Line 241: extra word - "on with"

Fixed.

Line 245: missing word – "models"

Fixed.